# Models of care and the role of clinical pharmacists in UK primary care for older adults: A scoping review protocol

**Nathan Davies**[1]*, **Vladimir Kolodin**[1], **Abi Woodward**[1], **Cini Bhanu**[1], **Yogini Jani**[2,3], **Jill Manthorpe**[4,5], **Mine Orlu**[6], **Kritika Samsi**[4,5], **Alice Burnand**[1,7], **Victoria Vickerstaff**[8], **Emily West**[1], **Jane Wilcock**[1], **Greta Rait**[1,8]

**1** Research Department of Primary Care and Population Health, Centre for Ageing Population Studies, University College London, London, United Kingdom, **2** Research Department of Practice and Policy, School of Pharmacy, University College London, London, United Kingdom, **3** Centre for Medicines Optimisation Research and Education, University College London Hospitals NHS Foundation Trust, London, United Kingdom, **4** NIHR Applied Research Collaborative (ARC) South London, King's College London, London, United Kingdom, **5** NIHR Policy Research Unit in Health & Social Care Workforce, King's College London, London, United Kingdom, **6** Research Department of Pharmaceutics, UCL School of Pharmacy, University College London, London, United Kingdom, **7** Department of Clinical and Movement Neurosciences, University College London, London, United Kingdom, **8** Research Department of Primary Care and Population Health, PRIMENT Clinical Trials Unit, University College London, London, United Kingdom

* n.m.davies@ucl.ac.uk

**Data Availability Statement:** No datasets were generated or analysed during the current study. All

## Abstract

### Introduction

There has been global investment of new ways of working to support workforce pressures, including investment in clinical pharmacists working in primary care by the NHS in the England. Clinical pharmacists are well suited to support older adults who have multiple long-term conditions and are on multiple medications. It is important to establish an evidence base for the role of clinical pharmacists in supporting older adults in primary care, to inform strategic and research priorities. The aim of this scoping review is to identify, map and describe existing research and policy/guidance on the role of clinical pharmacists in primary care supporting older adults, and the models of care they provide.

### Methods and analysis

A scoping review guided by the Joanne Briggs Institute methodology for scoping reviews, using a three-step strategy. We will search Medline, CINAHL, Scopus, EMBASE, Web of Science, PSYCHInfo, and Cochrane for English language articles, from 2015 –present day. Grey literature will be searched using Grey Matters guidelines, the Index of Grey Literature and Alternative Sources and Resources, and Google keyword searching. References of all included sources will be hand searched to identify further resources. Using the Population, Concept and Context framework for inclusion and exclusion criteria, articles will be independently screened by two reviewers. The inclusion and exclusion criteria will be refined after we become familiar with the search results, following the iterative nature of a scoping review. Data will be extracted using a data extraction tool using Microsoft Excel and presented using a narrative synthesis approach.

relevant data from this study will be made available upon study completion.

**Funding:** This study/project is funded by the National Institute for Health and Care Research (NIHR) School for Primary Care Research (project reference 575). The views expressed are those of the authors and not necessarily those of the NIHR or the Department of Health and Social Care. The funders had and will not have a role in study design, data collection and analysis, decision to publish, or preparation of the manuscript.

**Competing interests:** The authors have declared that no competing interests exist.

## Ethics and dissemination

Ethical approval is not required for this review. Review findings will be disseminated in academic conferences and used to inform subsequent qualitative research. Findings will be published and shared with relevant local and national organisations.

## Introduction

General practice and primary care are undergoing a workforce crisis globally [1,2]. Growing demand, an ageing population, and more people living with complex or multiple long-term conditions, including dementia, have changed the landscape of primary care. Workforce challenges in general practice; specifically general practitioner (GP) recruitment and retention, are compounding problems [1,3]. To meet these challenges and provide high quality care, a diversity of professions in primary care is needed. There is currently a broadening of skills and specialisms in primary care teams internationally, with care being delivered by multi-disciplinary teams including GPs, specialist nurses, occupational therapists, paramedics, pharmacists and social prescribers, among others [4,5].

Since the 1980s, pharmacists have increasingly provided direct patient care, and this has been reported in many countries including the United States, Australia, New Zealand, Malaysia, the Netherlands and the UK for example [6–11]. Their work within primary care has evolved to be more clinically orientated. Clinical pharmacists may work with community pharmacists, who provide patients with their medications. However, a clinical pharmacist will also conduct clinical assessments and prescribe treatment, care management of patients with long-term diseases, and clinical medication reviews to proactively help people with complex polypharmacy [10,12–14]. Clinical pharmacists also have a strategic role in implementing national health priorities at local level, maximising benefit whilst minimising risk associated with medicines and designing treatment pathways for patients [14]. NHS England defines a clinical pharmacist as "... *highly qualified experts in medicines and can help people in a range of ways. This includes carrying out structured medication reviews for patients with ongoing health problems and improving patient safety, outcomes and value through a person-centred approach.*" [15].

In 2015 there was substantial investment in pharmacists in the UK. NHS England launched the pilot of the Clinical Pharmacists in General Practice scheme (CPGP) and its associated training programmes, as part of the NHS Five Year Forward View [16] and revised arrangements for general practice contracts [17]. Similar programmes were also launched in the devolved nations of the UK during this period [18,19].

This CPGP pilot involved 490 clinical pharmacists being embedded across 658 GP practices aiming to ease GP workload and improve patient access. An independent evaluation of the pilot concluded clinical pharmacists have made a substantial contribution to the primary care skill mix, including contributing to patient safety, healthy lifestyles, and medication knowledge across the primary care team [20]. Subsequently, the government made a commitment to an additional 7,500 clinical pharmacists in primary care networks by 2024 [21,22]. Following the CPGP pilot two cross-sectional surveys identified a variety of enhanced responsibilities of clinical pharmacists in primary care including: medicine reconciliation, telephone support for patients, and face to face medication review [23,24]. Other studies focused on workforce elements including integration within the primary care and practice environment [12,25]. Although there is evidence of the benefit of community pharmacy led services and pharmacists

in hospital settings, there is a paucity of evidence within primary care, specifically related to clinical pharmacists [26,27].

Research on the benefits of having a clinical pharmacist in primary care for the general population have included identifying medicines-related problems, reduction in medication waste, and medicines optimisation [24,28]. Interviews with primary care teams and patients identified improved patient relationships, increased patient safety, enhanced cost savings and, importantly for patients, improved accessibility, with clinical pharmacists able to see patients much quicker than a GP or practice nurse [29–31].

The UK is seeing a rise in the ageing population, many of whom (67%) are living with two or more long term multiple conditions [32], including dementia and have associated high medication use and more rapid hospital discharge. Such factors have increased GPs' workloads [33]. There is currently no cure for dementia, so the focus is on symptom management, providing good quality care, with an emphasis on living well and quality of life. Medicines play a key role in the management of symptoms [34], coupled with 91.8% of people living with dementia also having another health condition [35]. The presence of multiple long-term conditions leads to complex management and the use of complex drug regimes, with many people taking multiple medications (polypharmacy) which require ongoing management and review. Medication errors and overprescribing are global challenges [11,36,37], and has been highlighted as a serious problem in the UK with physical and mental impacts on patients, hospital visits and preventable admissions, premature deaths and increased costs for the NHS [28]. Overprescribing may also disproportionately affect Black, Asian and ethnic minority communities in the UK [28,38]. The role of the clinical pharmacist in primary care and general practice has valuable potential for the wellbeing of older people.

As the numbers of clinical pharmacists expand in primary care, it is important to establish an evidence base for the role of clinical pharmacists in supporting older adults in primary care including those living with dementia, to inform strategic and research priorities. Importantly this review will highlight gaps in workforce planning, strategy, commissioning, and research to inform future work and priorities in research, policy, and practice. We will identify different models of care, by which we mean what services are provided, what they do and who employs the pharmacists.

A preliminary search of Prospero, the Cochrane Database of Systematic Reviews and Joanna Briggs Institute (JBI) Evidence Synthesis was conducted in 2022 and no systematic reviews or scoping reviews on the topic were identified, either published or underway.

## Research questions

The overarching aim of this review is to identify, map and describe existing knowledge and research on the role of clinical pharmacists in primary care supporting older adults, and the models of care.

**Primary review question.** What research has been conducted on clinical pharmacy services used in primary care for older adults (over the age of 65 years) in the UK?

## Secondary review questions

- What is the role of clinical pharmacists, and what models of clinical pharmacy are used in primary care for older adults?

- What are professional perceptions of the role of clinical pharmacists who are involved in the care of older adults in primary care?

- What are the experiences of older adults and their carers (family and friends) who have encountered clinical pharmacists in primary care?

- What are the experiences of older adults from minority ethnic groups and those from social deprivation?

- What is the role of clinical pharmacists, and what models of clinical pharmacy are used in primary care for people with dementia?

## Methods

### Design

The proposed scoping review will be conducted in accordance with the JBI methodology for scoping reviews [39,40]. The review will be guided by the PCC (Participant, concept, and context) framework as recommended in the JBI scoping review methodology, to inform our review questions, search strategy, and inclusion/exclusion criteria. This protocol is reported using the Preferred Reporting Items for Systematic Review and Meta-Analysis extension for scoping reviews (PRISMA-ScR) [41] and Preferred Reporting Items for Systematic Review and Meta-Analysis Protocols (PRISMA-P) checklist for systematic review protocols (see S1 Appendix) [42] adapted using guidance by JBI for best practice reporting of scoping review protocols [43].

### Eligibility criteria

An overview of eligibility criteria is provided in Table 1.

**Participants.** The population for this review is older people in primary care. In the UK the definition of an older adult is generally someone over the age of 65 years [44], and is the definition adopted for this review. We will also include papers which do not specify the age of participants but refer to them as 'older'. Sources which do not specify their focus is on older adults, or those over 65 years will not be included. Sources which have a mix of participants' ages will be included if they have a separate discussion or analysis on older adults. Sources

**Table 1. Overview of inclusion and exclusion criteria.**

|  | *Included* | *Excluded* |
|---|---|---|
| *Population* | Older people (65 years and above) or if the paper describes them as older adults/people | Mixed population with no separate analysis on older people |
|  | Professionals caring for older adults | Those under 65 years |
|  | Family and friends who are carers of older adults |  |
| *Concept* | Clinical pharmacist role | Community pharmacy |
|  | Clinical pharmacy in primary care |  |
|  | Focus on what clinical pharmacy services are delivered and how |  |
|  | Experiences of services from professionals, family carers and older adults' perspectives |  |
| *Context* | Primary care | Studies outside the UK |
|  | UK | Studies prior to 2015 |
|  | Published since 2015 and the introduction of the clinical pharmacist in general practice | Studies of the community pharmacy role |
|  | Care homes | Dentistry and optometry services |
|  |  | Non-English language |

which include professionals, older adults and carers will be included in the review. The definition of professionals will include doctors, pharmacists, practice nurses and other staff in primary care. The review will focus on unpaid carers who are a friend or family member of an older person who regularly takes care of the person due to their illness, frailty, or disability. Carers will be included with no restriction on age or other demographics.

**Concept.** Studies and resources that relate to clinical pharmacists who work in primary care will be included. Our definition of primary care is provided below in context.

We will include resources, including research, policy and grey literature, reporting: what services and how services are delivered, who is using services, the frequency with which services are being used, whether services are being responded to positively or otherwise. There are also likely to be resources on professionals' experiences of these services–both from the primary care and pharmacy perspective. Similarly, data may include patient and carer experiences in using (or not using) such services as well as resources, publications and reviews that explore the role of clinical pharmacists in primary care. In initial screening we will include condition specific sources (e.g. studies which focus on older adults with COPD), however we will review this decision in light of the results of abstract screening and refine our criteria as appropriate at that point. This is in line with the iterative nature of a scoping review (40).

**Context.** The context for this review is primary care, which is defined for this review as services which provide the first point of contact in the healthcare system [45], this includes but is not limited to general practice. For the purposes of this review, we will not include dental, optometry or community pharmacy services. However, we are aware clinical pharmacists may work in community teams and support community pharmacy services as part of their role employed by another primary care organisation (i.e. GP practice, primary care network, integrated care systems) and sources focussing on clinical pharmacists from those organisations will be included. We will include care delivered by clinical pharmacists to care homes. Resources and publications for this review will be from the UK context to account for specificities in service provision and the role of both primary care and pharmacy.

**Types of sources.** This scoping review will consider all primary research studies (qualitative and quantitative), systematic reviews, meta-analyses, letters to editors, commentaries, blogs and grey literature.

We will actively search for and include grey literature such as: theses, dissertations, trade publications, national policy and guidelines, reports, websites, conference posters, preprints, and others. This is relevant as this is a new emerging field, and more information may be published in these sources. This will minimise publication bias and maximise our understanding of the landscape. Non-English language studies and those published prior to 2015 will be excluded. We selected 2015 as the cut-off date for three reasons: 1) this was the start of the pilot of clinical pharmacists working in general practice and primary care in England [20]; 2) the timepoint of investment by the Scottish Government to support recruiting additional pharmacists to support the care of patients with long term conditions, freeing up GP time [18] and finally 3) the period when there significant investment in recruitment of clinical pharmacist in primary care teams in Wales [19]. However, we acknowledge that some parts of the UK may have other models of care which pre-date our cut-off date and this will be acknowledged in our review.

## Search strategy

The JBI recommends a three-step search strategy [40], firstly an initial limited search of two databases will be undertaken (CINAHL and MedLine) based on initial search terms developed from existing literature and discussions with an information specialist from the University

library. We will use a combination of keywords and Medical Subject Headings (MeSH) terms which will be adapted for the various databases, combined using Boolean operators ('AND' 'OR'). Results will be screened at title and abstract level to ensure sensitivity and specificity of the search strategy, ensuring that key articles already known to the research team are captured in the results. The search strategy will then be refined using search terms from identified articles in step one. For step two the revised search will be re-run in the initial two databases, as well as, Scopus, EMBASE, Web of Science, PSYCHInfo, and Cochrane (see S1 Appendix for example full search strategy). Grey literature will be searched guided by Grey Matters guidelines, the Index of Grey Literature and Alternative Sources and Resources and Google keyword searching. Following this, the final step will include hand searching the reference lists of included articles, contacting authors for further information if necessary and to identify any further sources, and finally we will conduct citation tracking using Google Scholar. We will work with our study patient and public involvement group (PPI) as well as key stakeholders to identify any additional sources our searches may not have identified.

Search terms will be identified prior to starting the review, however as the JBI Manual states the search for a scoping review may be iterative as reviewers become more familiar with the evidence base [40], we will therefore revise our search strategy as needed as we proceed with the review.

## Study/Source of evidence selection

Following the search, all identified citations will be collated and uploaded into Endnote, and duplicates removed. References will be imported into Rayann for screening [46]. Following a pilot test and initial screening to refine the search, titles and abstracts will then be screened by one reviewer (VK, AB, DL or ND) for assessment against the inclusion criteria for the review and a random 10% reviewed by a second reviewer (VK, AB, DL or ND). Potentially relevant sources will be retrieved in full. The full text of selected citations will be assessed in detail against the inclusion criteria by one reviewer (VK, AB, or ND) and 10% reviewed by a second reviewer (VK, AB, DL or ND). Reasons for exclusion of sources of evidence at full text that do not meet the inclusion criteria will be recorded and reported in the scoping review. Any disagreements that arise between the reviewers at each stage of the selection process will be resolved through discussion, or with an additional reviewer/s. The results of the search and the study inclusion process will be reported in full in the final scoping review and presented in a Preferred Reporting Items for Systematic Reviews and Meta-analyses extension for scoping review (PRISMA-ScR) flow diagram [41]. We will continually review our eligibility criteria and focus of the review based on the identified sources, and revise the eligibility criteria as needed in line with the iterative nature of scoping reviews [40]. We will not conduct quality assessment of included articles as this is not recommended by scoping review guidance [39,47].

## Data extraction

The data from the included sources will be extracted using a data extraction tool developed by the research team using Microsoft Excel. The tool will be tested on two sources initially to ensure all relevant information is being collected and modified as required. This will be conducted by two independent reviewers (VK or AB) and individual extraction compared, any disagreements that arise between the reviewers will be resolved through discussion, or with an additional reviewer/s. Following this the remaining included sources will be extracted by two independent reviewers (VK,AB, or AW). Throughout the review process the extraction tool will be modified as necessary as we learn more about the field and results, this will be detailed

in the review. If appropriate, authors of papers will be contacted to request missing or additional data, where required. The following information will be extracted:

- Author(s)
- Year of publication
- Country of origin
- Source type (i.e. journal article, report, website)
- Aims
- Study design (if applicable)
- Participant demographics
- Model of clinical pharmacy (i.e. services delivered, who employs the pharmacist, what they do)
- Clinical pharmacy role/definition
- Key findings qualitative
- Key findings quantitative
- Conclusions
- Strengths and limitations
- Implications for future research
- Implications for policy and practice
- Role of communication
- Context and special requirements when work with minority ethnic groups

## Data analysis and presentation

Findings will be presented separately for each of the research questions above. Initially, providing a taxonomy or figure to reflect the breadth and areas of research in this field. This will be followed by a series of tables and narrative summary of the included sources. Although scoping review guidance does not recommend the use of thematic analysis to analyse the data we will use the principles of codebook thematic analysis [48] to identify key themes across the sources and enable us to narratively present the remaining findings from the sources included. We will highlight gaps in the literature and identify future priorities for research, policy, and practice. The report will follow the PRISMA guidance for scoping reviews [41].

## PPI and stakeholder engagement

Our PPI group consisting of two family carers of older adults has been embedded in our research from the outset, advising on resources to consider and sources of information. We will present our emerging findings throughout the process to our PPI group and additional stakeholders including older people, family and friends who are carers, and professionals from within primary care, to inform our synthesis and identify gaps in the literature. During initial consolations, PPI representative suggested to focus on the role of communication between clinical pharmacist and patients, as well as specific approaches when working with people from minority ethnic groups.

## Impact

This scoping review will increase the understanding of the landscape of clinical pharmacy in primary care, specifically in relation to older adults. The information from the scoping review together with primary research to be conducted alongside the review will be presented in a series of stakeholder engagement workshops to produce a list of priorities for research. The outcome of this literature review will identify key services being delivered, patterns of engagement and utilisation of services, and perceptions of key stakeholders regarding these services. The review will provide evidence for clinical pharmacy services and older adults in primary care, directly influencing new and emerging agendas of the primary care workforce and the role of clinical pharmacists in the UK.

## Supporting information

**S1 Checklist. PRISMA-P 2015 checklist.**
(DOCX)

**S1 Appendix. Search strategy for MEDLINE.**
(DOCX)

## Acknowledgments

We would like to thank our PPI group for their involvement in the scoping review and development of this protocol. We would also like to thank David Green, an information specialist, who helped us devise the search strategy and advised on databases. We would like to thank Drew Leamon who has agreed to support with screening articles.

## Author Contributions

**Conceptualization:** Nathan Davies, Vladimir Kolodin, Cini Bhanu, Yogini Jani, Jill Manthorpe, Mine Orlu, Kritika Samsi, Victoria Vickerstaff, Emily West, Jane Wilcock, Greta Rait.

**Funding acquisition:** Nathan Davies, Cini Bhanu, Yogini Jani, Jill Manthorpe, Mine Orlu, Kritika Samsi, Victoria Vickerstaff, Emily West, Jane Wilcock, Greta Rait.

**Methodology:** Nathan Davies, Vladimir Kolodin, Abi Woodward, Cini Bhanu, Yogini Jani, Jill Manthorpe, Mine Orlu, Kritika Samsi, Alice Burnand, Victoria Vickerstaff, Emily West, Jane Wilcock, Greta Rait.

**Supervision:** Nathan Davies, Abi Woodward.

**Writing – original draft:** Nathan Davies, Vladimir Kolodin, Emily West.

**Writing – review & editing:** Nathan Davies, Vladimir Kolodin, Abi Woodward, Cini Bhanu, Yogini Jani, Jill Manthorpe, Mine Orlu, Kritika Samsi, Alice Burnand, Victoria Vickerstaff, Jane Wilcock, Greta Rait.

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
