## [Decision Letter · Decision Letter 0]

28 Mar 2023

PONE-D-22-27706

Models of care and the role of clinical pharmacists in UK primary care for older adults: A scoping review protocol

PLOS ONE

Dear Nathan,

Thank you for submitting your manuscript to PLOS ONE. After careful consideration, we feel that it has merit but does not fully meet PLOS ONE’s publication criteria as it currently stands. Therefore, we invite you to submit a revised version of the manuscript that addresses the points raised during the review process.

Please submit your revised manuscript by 24/04/2023. If you will need more time than this to complete your revisions, please reply to this message or contact the journal office at plosone@plos.org. Please include the following items when submitting your revised manuscript:

We look forward to receiving your revised manuscript.

Kind regards,

Surangi Jayakody, MBBS, MSc, MD

Academic Editor

PLOS ONE

Reviewers' comments:

Reviewer's Responses to Questions

**Comments to the Author**

1. Does the manuscript provide a valid rationale for the proposed study, with clearly identified and justified research questions?

Reviewer #1: Yes

Reviewer #2: Yes

2. Is the protocol technically sound and planned in a manner that will lead to a meaningful outcome and allow testing the stated hypotheses?

Reviewer #1: Yes

Reviewer #2: Partly

3. Is the methodology feasible and described in sufficient detail to allow the work to be replicable?

Reviewer #1: No

Reviewer #2: Yes

4. Have the authors described where all data underlying the findings will be made available when the study is complete?

Reviewer #1: Yes

Reviewer #2: Yes

5. Is the manuscript presented in an intelligible fashion and written in standard English?

Reviewer #1: Yes

Reviewer #2: Yes

6. Review Comments to the Author

You may also provide optional suggestions and comments to authors that they might find helpful in planning their study.

Reviewer #1: There are few concerns which need to be cleared before publications and they are given as the comments.

Reviewer #2: Thank you for the opportunity to review this interesting paper. I think that with the following amendments, this paper would be suitable for publication in PLOS ONE.

Abstract

Methods: the date (month/year- from 2015) the search will be conducted and any search limits (e.g. language- Only English language articles) need to be added.

Methods

Data Extraction

Page 13- Following this the remaining included sources will be extracted by 304 one reviewer (VK or AB).

Data need to be extracted from papers included in the scoping review by two or more independent reviewers using a data extraction tool.

7. PLOS authors have the option to publish the peer review history of their article (what does this mean?). If published, this will include your full peer review and any attached files.

Reviewer #1: No

Reviewer #2: No

---

## [Author Response · Author response to Decision Letter 0]

3 Apr 2023

We would like to thank both reviewers for their positive and helpful comments which we feel have strengthened our manuscript. All page numbers and line numbers refer to the tracked changes document. 

Reviewer #1: There are few concerns which need to be cleared before publications and they are given as the comments.

Interesting and timely topic to explore.

Thank you we agree this is a very timely topic. 

It would be better if the pre-determined tool can be mentioned.

It is not a tool which already exists but one which we will develop as a team – we have clarified this in the abstract and then on page 13 in the data extraction section we believe this is clearer. On page 13-14 we specify what is included in the data extraction tool/template.

Is this "older adults"? (page 7 line 174)

Yes, we have added older to this question.

It is better to add in the introduction why dementia is taken out of all the multi morbidities in elders. 

We have added information on dementia on page 6 lines 131-136 and 149 to introduce this before the aims.

Why are the unpaid carers are excluded? 

Unpaid carers (i.e. family and friends) are included – this is discussed on page 8 line 206-208 and again in the eligibility criteria table on page 11.

Are these consensus documents? (Page 10, line 239)

No we do not think these are consensus documents. 

what about paid carers? (inclusion criteria)

We are including professionals caring for older adults, this is in our eligibility criteria table on page 11. 

Reviewer #2: Thank you for the opportunity to review this interesting paper. I think that with the following amendments, this paper would be suitable for publication in PLOS ONE.

Abstract

Methods: the date (month/year- from 2015) the search will be conducted and any search limits (e.g. language- Only English language articles) need to be added.

We have added 2015-present day as it is not possible to specify a date to confirm when the final search will be completed as we would update this before publication of the finalised review. However, we will specify this in the main article when publishing. We have added the limit English language. 

Methods

Data Extraction

Page 13- Following this the remaining included sources will be extracted by one reviewer (VK or AB).

Data need to be extracted from papers included in the scoping review by two or more independent reviewers using a data extraction tool.

Thank you for raising this we have amended the protocol see page 13 line 313.

---

## [Decision Letter · Decision Letter 1]

8 Jun 2023

Models of care and the role of clinical pharmacists in UK primary care for older adults: A scoping review protocol

PONE-D-22-27706R1

Dear,

We’re pleased to inform you that your manuscript has been judged scientifically suitable for publication and will be formally accepted for publication once it meets all outstanding technical requirements.

Kind regards,

Muhammad Shahzad Aslam, Ph.D.,M.Phil., Pharm-D

Academic Editor

PLOS ONE

Additional Editor Comments (optional):

Reviewers' comments:

Reviewer's Responses to Questions

**Comments to the Author**

1. Does the manuscript provide a valid rationale for the proposed study, with clearly identified and justified research questions?

Reviewer #3: Yes

2. Is the protocol technically sound and planned in a manner that will lead to a meaningful outcome and allow testing the stated hypotheses?

Reviewer #3: Yes

3. Is the methodology feasible and described in sufficient detail to allow the work to be replicable?

Reviewer #3: Yes

4. Have the authors described where all data underlying the findings will be made available when the study is complete?

Reviewer #3: No

5. Is the manuscript presented in an intelligible fashion and written in standard English?

Reviewer #3: Yes

6. Review Comments to the Author

You may also provide optional suggestions and comments to authors that they might find helpful in planning their study.

Reviewer #3: In my opinion, the initial concerns raised had been addressed by the authors in this version.

Suggest to add clearly where the data underlying the findings will be made available when the study is complete.

7. PLOS authors have the option to publish the peer review history of their article (what does this mean?). If published, this will include your full peer review and any attached files.

Reviewer #3: **Yes: **Dr Mathumalar Loganathan

---

## [Editor Report · Acceptance letter]

18 Jul 2023

PONE-D-22-27706R1 

Models of care and the role of clinical pharmacists in UK primary care for older adults: A scoping review protocol 

Dear Dr. Davies:

I'm pleased to inform you that your manuscript has been deemed suitable for publication in PLOS ONE. Congratulations! Your manuscript is now with our production department. 

Kind regards, 

on behalf of

Dr. Muhammad Shahzad Aslam 

Academic Editor

PLOS ONE